# Construction of a Prognostic Score for Ultrasound Evaluation of the Transobturator Sling for Stress Urinary Incontinence

**DOI:** 10.3390/jcm11051296

**Published:** 2022-02-27

**Authors:** Espada-Gonzalez Cristina, Sabonet-Morente Lorena, Perez-Gonzalez Rita, Gonzalez-Mesa Ernesto Santiago, Jimenez-Lopez Jesus Salvador

**Affiliations:** 1Department of Gynecology and Obstetrics, Faculty of Medicine, University of Málaga, 29010 Malaga, Spain; doctoracristinaespada@gmail.com (E.-G.C.); egonzalezmesa@gmail.com (G.-M.E.S.); jesuss.jimenez.sspa@juntadeandalucia.es (J.-L.J.S.); 2Department of Gynecology and Obstetrics, Quiron Salud Hospital of Malaga, 29004 Malaga, Spain; 3Department of Gynecology and Obstetrics, Regional Universitary Hospital of Malaga, 29011 Malaga, Spain; 4Statistics Department, The Institute of Biomedical Research of Málaga (IBIMA), Regional Universitary Hospital of Malaga, 29011 Malaga, Spain; rita.perez@ibima.eu

**Keywords:** stress urine incontinence, anti-incontinence sling, anti-incontinence surgery, pelvic floor ultrasound, prognostic score

## Abstract

Currently, pelvic floor ultrasound allows us to correctly visualize the synthetic material used in stress urinary incontinence surgery. The objective of this study is the construction of a score and its correlation with the SUU clinic. During the study period, 81 patients with transobturator slings were studied using ultrasound. Through multivariate analysis, the statistically significant variables were the distance from the sling to the urethral wall (*p* = 0.004), the shape of the sling at rest (*p* = 0.003), and the symmetry of the mesh (*p* = 0.016). Through these variables, the construction of a score was carried out. Once the model was constructed, its internal validation was carried out to determine the discrimination capacity of patients who present clinical stress and those who do not, with an area under the curve of 0.848 (95% CI (0.72–0.97), *p* < 0.001). This simple score using three ultrasound variables serves to adequately and objectively discriminate patients who have successful surgery and absence of clinical effort.

## 1. Introduction

Urinary incontinence (UI) is a very frequent pathology with a prevalence of 37.1% in the world’s female population that can manifest at any age, increasing its frequency after the age of 50, seriously affecting the quality of life for patients [1].

In 2010, the International Urogynecological Association (IUGA)/International Conti-nence Society (ICS) published the standardization of terms defining urine incontinence (UI) as any involuntary loss of urine [2,3]. UI can be classified into three types: stress, ur-gency, or mixed. Stress urinary incontinence (SUI) is in response to increased in-tra-abdominal pressure, such as from coughing, laughing, sneezing, exercising, or weightlifting. Urge incontinence is the involuntary loss of urine associated with an urge for urination that is difficult to control. Mixed-type urine incontinence consists of a com-bination of the above types.

Stress urinary incontinence (SUI) in the female population is mainly caused by the loss of muscle or ligamentous support that generates an inadequate coaptation of the urethra, combined with an increase in intra-abdominal pressure. This generates a displacement of the urethra, and the lower edge of the bladder moves downwards, which is defined as urethral hypermobility [4].

The most common surgical technique used for the correction of stress urinary incontinence is based on the restoration of the urethral support and correction of the deficient urethral closure by implanting suburethral support or transobturator sling made of synthetic material in the middle urethra, without tension, through three minimal incisions, one of them vaginal and two inguinal [5,6,7]. These techniques use transobturators and suburethral tension-free slings. Currently, the most used technique is the transobturator sling described by Delorme in 2001 for the reduction of bladder and urethral damage assessed with suburethral tapes, due to the absence of the need for posterior cystoscopy [8,9].

Pelvic floor ultrasound is considered the best imaging technique for the assessment of tapes and synthetic material [10,11]. It also provides a good correlation with surgical exploration since tapes and materials are easily observed when generating a hyperechogenic image, facilitating assessments of post-surgical complications. Pelvic floor ultrasound assesses the urethral path, urethral hypermobility, and post-surgical changes in anti-incontinence surgeries by visualizing the slings [12,13]. Being a dynamic test allows physicians to assess the sling at rest and in Valsalva, and thus to determine its functionality and observe post-surgical complications such as surgical obstruction, the presence of hematomas, extrusion of the mesh, or the presence of other concomitant problems such as pelvic organ prolapses.

Urethral hypermobility can be assessed in an ultrasound way through the mediosagital plane by measuring the urethral length from the bladder neck to the symphysis pubis when this distance is shortened in Valsalva. Urethral hypermobility is a result of the loss of the urethral support and the descent of the bladder neck, although there is no consensus on the cut-off point in the literature [14] (see Figure 1). In addition, it can be determined by measuring the posterior retrovesical angle formed by the path of the urethra and the bladder trine, which is increased in the Valsalva because of this descent of the bladder neck [15].

The anti-incontinence sling parameters assessable by ultrasound are [16,17]:Position of the sling with respect to the urethra (proximal, middle, or distal third) both at rest and in Valsalva.Distance between the sling and the posterior urethral wall is considered adequate if between 3–5 mm (in 2D transperineal). With a high-frequency probe, it is possible to accurately calculate the distance between the sling and the urethral light.Sling shape (at rest and Valsalva). The tapes usually have a flat shape at rest, determined by their shape parallel to the urethral lumen, and they deform into a “C” shape in Valsalva in relation to the tension caused by the increase in intra-abdominal pressure.Concordance of the urethral movement with that of the sling during the Valsalva (urethral kinking).The symmetry of the sling and its arms in the cross-section or corona. It allows assessment of the extrusion or winding of the sling arms.

Slings in the proximal urethral third usually give irritative and obstructive symptoms, as do those that are too close to the urethral mucosa. Wrinkled slings usually slide into the Valsalva without pinching the middle urethra and suppose the failure of the technique with persistence or recurrence of incontinence; those located too distally are usually less effective [12,16].

The objective of this study is the construction of a score through ultrasound parameters and its correlation with the clinical effort to be able to objectively and easily evaluate the success of the anti-incontinence surgical technique.

## 2. Materials and Methods

The study was conducted in accordance with the Declaration of Helsinki of the World Health Organization. The study protocol was approved by the Ethics Committee of the Regional University Hospital of Malaga (2594-N-20 dated 2021). All patients included in the study were correctly informed of the entire process and subsequently signed the informed consent.

The study population was female patients over 18 years of age who underwent transobsturatric mesh surgery in the health area of the Regional University Hospital of Malaga in the period from May 2018 to May 2020. The exclusion criteria were the lack of adequate follow-up visits, patients operated on surgically in another center, and patients with pregnancy after surgery. The total sample of patients who were included in the study was 85, of which 4 were lost due to inadequate follow-up visits made.

The face-to-face visit was carried out between 6 and 8 months after anti-incontinence surgery using an established protocol that included a complete medical history, history of the current clinic, quality of life questionnaire (ICIQ-UI-SF) [18], a physical examination, and an ultrasound evaluation. The current symptomatology was defined by the current terminology and definition of ICS: the presence or absence of involuntary loss of urine associated with symptoms of urgency, stress, or mixed (combination of both) [3].

The physical examination was carried out in the lithotomy position by assessing the clinical stress urinary incontinence through the Valsalva stress test with bladder repletion after 2 h without urination, the presence of pelvic organ prolapses, and the examination of the vagina to rule out extrusion of the mesh.

All ultrasounds were performed by the same operator using Voluson E8 (General Electric Healthcare) ultrasound in Málaga, Spain, equipment with a RAB6-D (2–8 MHz) volumetric convex transducer through the transverse and sagittal plane of the translabial pathway. In the second time, the measures and angles were verified by a second observer analyzing the iconography.

The ultrasound parameters evaluated were urethral length at rest; Valsalva, posterior urethrovesical angle at rest; Valsalva (PUVA), the shape of the sling at rest; Valsalva, minimum distance from the sling to the posterior urethral wall (MDPUW), symmetry of the sling, the position of the sling and concordance of the sling to the Valsalva assessed by the existence of slippage of the tape with the Valsalva.

The variable success of the surgery was obtained as the absence of clinical stress urinary incontinence. Initial construction and development of the score were carried out by constructing a logistic regression model by steps, forward. Considering as a dependent variable the success of the surgery and as independent variables different ultrasound variables, variables with a *p*-value < 0.05 were considered statistically significant. Subsequently, once the model was built, the performance of the model was assessed by evaluating the parameters related to calibration and discrimination capacity. The calibration of the model was assessed with the calibration belt, a graphical approach designed to evaluate the goodness of fit in a logistic regression model [19]. Statistical analysis was performed through the statistical software R Project, version 4.1.0.

## 3. Results

### 3.1. Descriptive Study

The study population that completed follow-up was 81 patients, with a mean age of 51.71 years and an age range of 35 to 76 years. The time elapsed until the post-surgical visit was 6.87 months. Among the risk factors analyzed (see Table 1), obesity with BMI > 28 was present in 36 patients (44%) and a history of depression or anxiety was present in 18 of the 81 patients (22%).

The total number of Assisted Vaginal Delivery was 189 deliveries, with an average parity of 2.33 + 1.16 vaginal deliveries. Only 2 (2.47%) patients had births by caesarean section. The presence of more than 1 vaginal birth was 62 (76.54%) patients, and deliveries of newborns weighing above 4000 g were present in 12 (14.81%) patients.

The presurgical clinic saw clinical symptomatic SUI in 52 (64.2%) patients and mixed symptoms in 29 (35.8%) patients. After surgery, the presence of clinical stress urinary incontinence was found in 6 (7.41%), urge symptoms in 16 (19.75%), mixed symptoms in 10 (12.35%), and total absence of symptoms in 49 (60.75%) patients. We considered the success of anti-incontinence surgery in 65 (80%) patients with the absence of symptomatic SUI.

### 3.2. Bivariate Analysis of the Sonographic Parameters

The association of different ultrasound parameters with the presence of clinical stress urinary incontinence was analyzed, finding a statistically significant association with the following variables: distance from the sing to the posterior urethral wall less than 5 mm (*p* < 0.001), the position of the sling in the distal or middle zone (*p* = 0.04), the flat shape of the sling at rest (*p* < 0.001), the symmetry of the sling (*p* = 0.008) and the presence of urethral concordance (*p* = 0.04) (Table 2).

### 3.3. Predictive Model

With the sample of 81 patients, a score was constructed through a logistic regression model. The obtained results determined to be statistically significant variables for the construction of a score for the clinical assessment of the transobturator sling were: the symmetry of the sling with an OR of 6.95 (95% CI: 1.48–36.96) (*p* = 0.016), the shape of the sling at rest flat with an OR of 11.63 (95% CI: 2.41–66.24) (*p* = 0.003) and the distance of the sling to the posterior urethral wall less than 5 mm with an OR of 10.84 (95% CI: 2.20–63.11) (*p* = 0.004) (Table 3).

The score consists of 3 variables with a total score of 0 to 3 points, where 0–1 point corresponds to the presence of a clinical SUI and therefore we can consider the non-success of the surgical technique and 2–3 points that determines the absence of clinical stress uri-nary incontinence and with it the success of the surgery.

With the score of 0–1 point we are able to discriminate the presence of clinical stress urinary incontinence by 56.2% in our sample and with 2–3 points we can categorize 96.9% of our sample that present absence of clinical stress urinary incontinence (see Table 4).

To evaluate the performance of the model, its parameters of calibration and discrimination were obtained.

In Figure 2, we can observe that both the 80% and 95% calibration belts encompass the bisector over the whole range of the predicted probabilities. We obtained a *p*-value = 0.21. These results suggest that the hypothesis of good calibration was not rejected so, the model’s internal calibration is acceptable. The area under the curve ROC (*p* < 0.001) was 0.848 with a CI of 95% (0.72–0.97), indicating the good capacity of the model to discriminate among the success and failure of the surgery (Figure 3).

## 4. Discussion

Our prospective study using the ultrasound parameters showed statistically significant results for the prediction of clinical stress urinary incontinence in patients who were operated on in the transobsturator with tension-free slings. This allows us to objectively identify patients with an absence of clinical effort and, therefore, to determine the success of the surgery in a simple and accessible way. 

The post-surgical assessment of anti-incontinence slings is normally evaluated through clinical signs and symptoms and quality of life questionnaires, in combination with isolated ultrasound parameters, but no standardized imaging technique has been described for their assessment [20,21].

The assessment of the symmetry of the sling allows the assessment of both arms significantly (*p* < 0.0001) according to our data, as well as the data of Illiano et al. [22], that considered incontinent those that presented with asymmetry in one of the arms.

In the literature, there is no unanimity in the measurement of the distance from the sling to the posterior urethral wall. Wermuth et al., in 2021 [17], in an attempt to unify these criteria, published a guide for the ultrasound assessment of suburethral slings, but continues to leave freedom for each institution to determine the extent of this parameter, some authors perform the measurement from the symphysis of the pubis [12], while others take the urethral wall as a reference. In the case of our study, we established the correct measurement as the minimum existing measurement between the edge of the sling and the posterior urethral wall and considered the cut-off point of 5 mm, according to the current literature [23,24], to discriminate the presence of symptomatic SUI, being the main objective of the surgery performed. In the case of slings very close to the urethral lumen, if there is consensus in the literature of their relationship with urge symptoms [25], these symptoms must be considered, and an ultrasound study was completed in search of these findings.

Kociszewski et al. [23], established an ultrasound classification according to the behavior of the sling according to its shape and considered that those that have a curved shape maintained at rest and in Valsalva to be related to greater complication and unsatisfactory surgery [24,25]. In our case, the correlation in Valsalva was not statistically significant, and therefore it was not included in the score. However, the flat form at rest was significantly related to the absence of clinical stress urinary incontinence, with p = 0.0003.

Other ultrasound parameters widely studied in the literature are the presence of ure-thral hypermobility directly related to the clinical SUI, ecographically assessable through the decrease in urethral length in Valsalva, and the increase to the posterior urethrovesical angle [14,26,27]. In our study, they were not statistically significant for the post-surgical assessment and therefore were not included in the score.

The position of the resting sling in the urethral path was also a statistically significant result (*p* = 0.04) in our study. The fact of performing a bivariate study (proximal versus non-proximal) may not be the most appropriate, since slings very far from the middle third are also related to the failure of the surgery [23,25]. The concordance of the movement of the sling with the urethra when performing Valsalva was also statistically significant (*p* = 0.04). Like other studies, we also consider it to be a useful parameter when the sling remains fixed at the middle position of the urethra [16,25]. These last ultrasound parameters (resting mesh position and urethral concordance) were not included in our score since they were not statistically significant. 

There are other parameters not included in our study, such as those analyzed by the group of Tan et al., which have also been shown to predict the results of anti-incontinence surgery, such as the assessment of the distance from the sling to the symphysis of the pubis when it is at 10–12 mm, and the angle in Valsalva described between the symphysis of the pubis and the edges of the sling being between 45–85° [28]. This study shows promising results, but they have a very limited sample size as it only includes 50% of the population.

The limitation of our study is the limited sample size and the absence of an external sample against which to perform external validation.

## 5. Conclusions

In summary, according to the current literature, this study offers the first prognostic score for the assessment of the post-surgical results of anti-incontinence slings through the exclusive use of ultrasound parameters. The great reproducibility and access to this score through a two-dimensional ultrasound allows us to objectively obtain a prognostic method that helps us in our daily practice of the post-surgical assessment of transobturator tension-free slings.

## Figures and Tables

**Figure 1 jcm-11-01296-f001:**
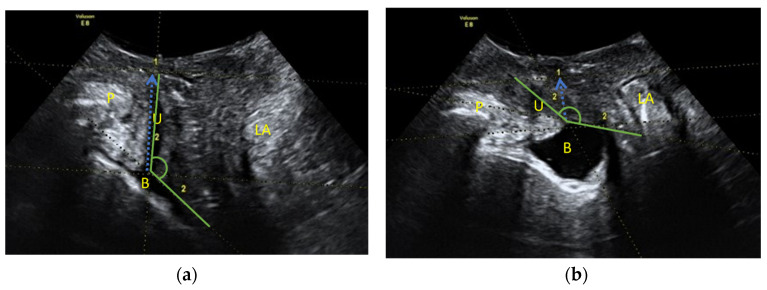
Sagittal median plane of pelvic floor ultrasound. We observe from the left to the right: symphysis pubis (S), urethra and bladder (U and B), vaginal area, rectum (R) and levator anis muscle (LE). Note in the posterior area of the urethra a hyperrefringent image compatible with the transobturator tension-free sling. (**a**) Resting image of the sagittal middle plane: (blue arrow) measurement of the urethral length and (green angle) the posterior urethrovesical angle. (**b**) Valsalva image of the sagittal middle plane: (blue arrow) measurement of urethral length and (green angle) posterior urethrovesical angle.

**Figure 2 jcm-11-01296-f002:**
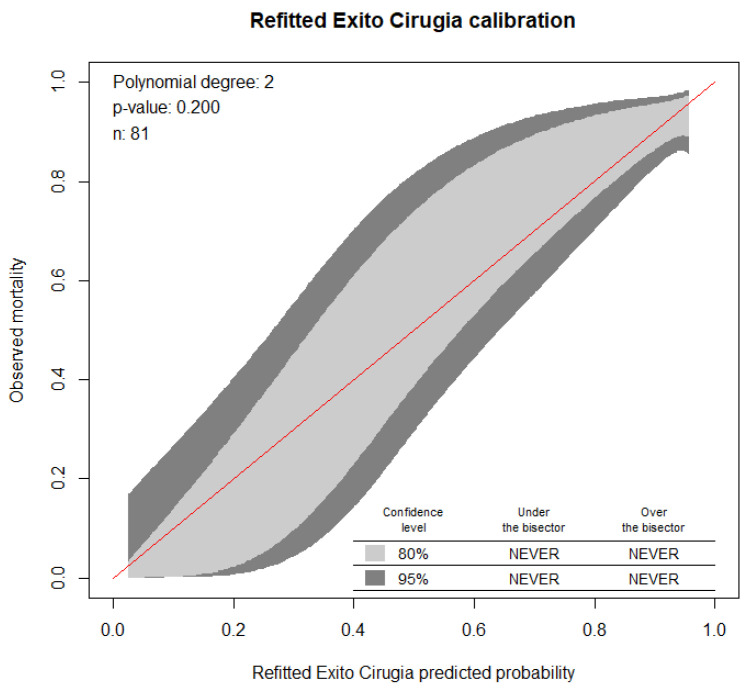
Calibration belt plot of the model. This figure shows the probability of predicting the success of the surgery through the score with an adequate calibration capacity (*p* = 0.200) with a confidence level of 95%, for which the study hypothesis is not rejected.

**Figure 3 jcm-11-01296-f003:**
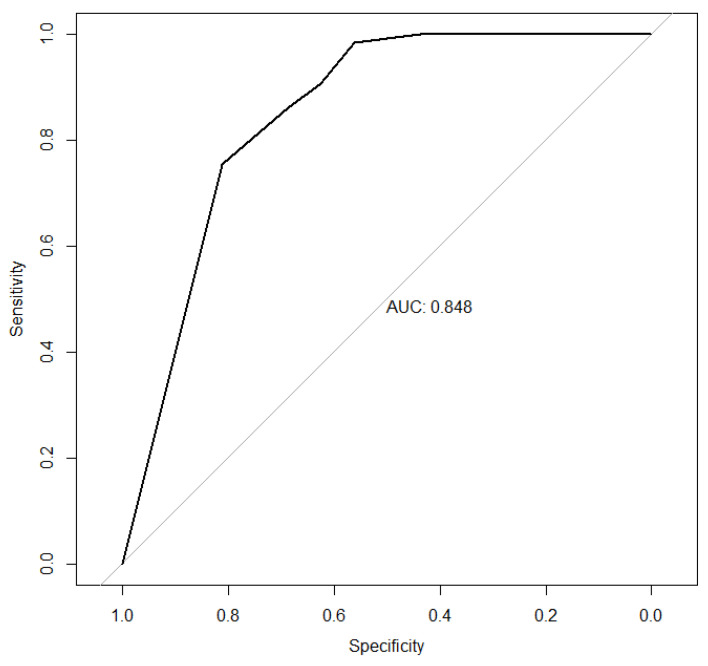
This figure shows the area under the curve of the discrimination capacity of the score developed for the absence of clinical effort.

**Table 1 jcm-11-01296-t001:** Descriptive study of the demographic characteristics of the patients studied. N = 81 patients operated on with transobturator tension-free sling.

Parameter	Media ± S.D.	n/N y%
Age	51.71 ± 10.57	
Obesity		36/81 (44%)
History of depression/anxiety		18/81 (22%)
Parity	2.33 ± 1.16	
Multiparity		62/81 (76.54%)
Childbirth >4000 g		12/81 (14.81%)
Cesarean deliveries		2/81 (2.47%)
Pre-surgical clinical		
Stress UI		52/81 (64.2%)
Mixed UI		29/81 (35.8%)
No clinic		0/81 (0%)
Post-Surgical clinical		
Stress UI		6/81 (7.41%)
Mixed UI		10/81 (12.35%)
Urgency		16/81 (19.75%)
No clinic		49/81 (60.49%)
Post-surgery ICIQ-IU	11.13 ± 11.79	

**Table 2 jcm-11-01296-t002:** Ultrasound parameters and their correlation with clinical stress. N = 81 patients.

Ultrasound Parameter	Value	Non-Clinical Stress UI (n = 33)	Clinical Stress UI (n = 9)	Total
Distance to posterior urethral wall (*p* < 0.001)	<5 mm	61 (93.8%)	8 (50%)	69 (85.2%)
	>5 mm	4 (6.2%)	8 (50%)	12 (14.8%)
Position at rest (*p* = 0.04)	Not proximal	65 (100%)	14 (87.5%)	79 (97.5%)
	Proximal	0 (0%)	2 (12.5%)	2 (2.5%)
Form at rest (*p* < 0.001)	In C	5 (7.7%)	8 (50%)	13 (16.1%)
	Flat	60 (92.3%)	8 (50%)	68 (84%)
Form in Valsalva (*p* = 0.24)	In C	57 (87.7%)	12 (75%)	69 (85.2%)
	Flat	8 (12.3%)	4 (25%)	12 (14.8%)
Resting length—Valsalva (*p* = 0.17)	<15 mm	63 (96.9%)	14 (87.5%)	77 (95.1%)
	15–25 mm	2 (3.1%)	2 (12.5%)	4 (4.9%)
Increased posterior urethrovesical angle (Valsalva—rest) (*p* = 0.21)	<15°	58 (89.2%)	12 (75%)	70 (86.4%)
	>15°	7 (10.8%)	4 (25%)	11 (13.6%)
Symmetry (*p* = 0.0083)	Asymmetric	8 (12.3%)	7 (43.8%)	15 (18.5%)
	Symmetrical	57 (87.7%)	9 (56.2%)	66 (81.5%)
Urethral concordance (*p* = 0.04)	Yes	65 (100%)	14 (87.5%)	79 (97.5%)
	No	0 (0%)	2 (12.5%)	2 (2.5%)

**Table 3 jcm-11-01296-t003:** Ultrasound parameters and statistically significant results for the construction of the score.

Parameter	OR	IC 95%	*p*-Value
Distance sling to posterior urethral wall < 5 mm	10.84	2.20–63.11	0.004
Flat shape of the sling at rest	11.63	2.41–66.24	0.003
Sling symmetry	6.95	1.48–36.96	0.016

**Table 4 jcm-11-01296-t004:** In this table we compare the discrimination capacity of the score developed to classify patients according to the presence of clinical stress or its absence.

		Clinical Stress UI	Non-Clinical Stress UI	Total
Clinic	YES	16 (100%)	0 (0%)	16 (19.8%)
	NO	0 (0%)	65 (100%)	65 (80.2%)
Score	0–1 points	9 (56.2%)	2 (3.1%)	11 (13.6%)
	2–3 points	7 (43.8%)	63 (96.9%)	81 (100%)

## Data Availability

The datasets used and analysed during the current study are available from the corresponding author on reasonable request.

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
