# Peer review of "Construction of a Prognostic Score for Ultrasound Evaluation of the Transobturator Sling for Stress Urinary Incontinence"

_jcm, 2022, doi:10.3390/jcm11051296_

Round 1

Reviewer 1 Report

Please change figures description to english.

I don t think it s first score for SI.

Author Response

ANSWER REVIEWER 1

As a response to the reviewer's comment we change to English in table 2 and 4.

The manuscript has been edited English language on December 2021 through MDPI. Attached certificate of the same which presents an erroneous title by a modification ultimately in agreement with all the authors of the manuscript.

Although we agree with the referee that this score is not the first score to evaluate stress urine incontinence, this score is the first to correlationate the transperineal ultrasound with the stress clinical

Reviewer 2 Report

With great interest I read the article written by ms Espada-Gonzalez et al about a prognostic score for ultrasound evaluation after SUI. The study presented is well performed and clearly written. Some minor grammatical errors need to be corrected and a spell check needs to be performed. 

This study nicely shows how a simple score, based on ultrasound evaluation, can determine the chance of succes after SUI surgery. To me it looks like a worthy tool in some cases. Nevertheless, some comments can be made:

  1. For me the clinical relevance of the ultrasound evaluation and score remains a bit unclear. In our daily clinic we evaluate the succes after SUI by complaints and satisfaction (also with questionnaires). I think this should be the main focus for possible subsequent actions (re-do surgery or medication). What does the ultrasound actually add to these subsequent actions? Does it have any clinical consequences. If patients tell you they are still incontinent, why do you need the ultrasound? Because to me it looks like an extra 5-10 min work that has no clinical consequences in a daily practice. I would like to encourage the authors to comment on this and to clarify this in the manuscript.
  2. The measurements are performed by only one observer. This is a good thing for consistency of the measurements in their local patient population, but how about inter-observer variability? 
  3. The authors do not describe the use of anticholinergics before and especially after surgery (Urge occurs in 20% of patients). Does urgency become less with simple medication?

Author Response

ANSWER REVIEWER 2

Comment 1: Through transperineal ultrasound and associating it with the clinic we can assess in an appropriate and simple way the need for a reoperation for a second sling when we obtain a low score and the presence of a stress clinic, which means that the sling is not in the right position and therefore does not anatomically correct the defect. In complex cases that do not meet all the criteria if we can opt for other surgical techniques such as TVT to try to correct the anatomical defect.

On the other hand, if we observe urgency we can see what the defect is, sling very close to the urethral lumen or bladder neck and susceptible to removal of the same if the symptoms are severe. Or if we observe that the sling is not symmetrical and is associated with extrusion of the sling in the vagina we can locate exactly where the defect is for correction.

In conclusion, it is an imaging test that gives us one more tool in cases of persistence of stress clinical and to be able to support us in our decision of a second anti-incontinence sling.

Comment 2: The referee is right to point out inter-observer variability, yet the measurements were initially made in a study of 30 patients by 2 observers with no significant differences between them. Given these findings, the present study was simplified into only 1 observer and in the second time checked by a second observer analyzing the iconography of tests and measurements. 

Comment 3: We agree that this is an important area that requires further research. In the present study we performed a clinical and ultrasound assessment of the clinical stress which is the one that has been analyzed in the score. We assess the anatomical correction of the sling and the disappearance of the clinical stress, being included as a success variable in the prediction score. On the other hand, the presence of urgent clinic is associated with inadequate slings or very close to the urethral lumen causing an irritation. By means of ultrasound we can assess and suspect this clinic if the sling is less than 3 mm from the posterior urethral wall or a sling very close to the bladder neck. In future studies it would be appropriate to check the clinical rate of urgency and correlate it with the ultrasound findings as well as its clinical evolution when applying anticholinergic treatment.

Reviewer 3 Report

  1. Extensive editing of English language and style required
  2. In table 2 and 4 you have used Spanish language!
  3. ACOG - Assisted Vaginal Delivery, you have vaginally assisted deliveries
  4. The term " suburethral band" is rather rarely used. You should change it for " suburethral sling"
  5. The term " Non-Clinical Stress" and " Clinical Stress" look moore like psychological terms, change it for " Non-Clinical Stress UI" and " Clinical Stress UI"

Author Response

ANSWER REVIEWER 3

Comment 1: The manuscript was edited in English in December 2021 through MDPI. Attached is a certificate of the same that presents an erroneous title due to a definitive modification in agreement with all the authors of the manuscript.

Comment 2: As a response to the reviewer's comment we change to English in table 2 and 4.

Comment 3: In accordance with the referees' wishes, we have now changed the term to "Assisted Vaginal Delivery".

Comment 4: In accordance with the referees' wishes, we have now changed those terms to "suburethral sling".

Comment 5: In accordance with the referees' wishes, we have now changed this sentence to " Non-Clinical Stress UI" and " Clinical Stress UI" from table 2 and 4.

This manuscript is a resubmission of an earlier submission. The following is a list of the peer review reports and author responses from that submission.